# The Effect of Work Stress on the Well-Being of Primary and Secondary School Teachers in China

**DOI:** 10.3390/ijerph20021154

**Published:** 2023-01-09

**Authors:** Jingyi Liao, Xin-Qiang Wang, Xiang Wang

**Affiliations:** 1Medical Psychological Center, The Second Xiangya Hospital of Central South University, Changsha 410011, China; 2Center of Mental Health Education and Research, School of Psychology, Jiangxi Normal University, Nanchang 330022, China

**Keywords:** primary and secondary school teachers, well-being, work stress, family–work conflict, self-transcendence meaning of life

## Abstract

Primary and secondary school teachers face increasing work stress, and more attention needs to be paid to their well-being. The present study was conducted to analyze the influence of work stress on the well-being of such teachers in China, and to explore the effects of family–work conflict and a self-transcendent meaning of life. A total of 562 primary and secondary school teachers completed questionnaires assessing work stress, family–work conflict, and a self-transcendent meaning of life (including grasping the meaning of failure and detachment from success or failure) as potential predictors of well-being. Work stress negatively predicted teachers’ well-being; family–work conflict mediated this relationship and a self-transcendent meaning of life moderated it. The results of this study can be used as a reference for education departments seeking to intervene to prevent teachers from developing well-being problems from the perspective of a self-transcendent meaning of life.

## 1. Introduction

Primary and secondary school teachers are facing increasing challenges and pressures with education reform and high expectations from society, schools, students, parents, and others. Although the well-being of these teachers differs among regions, most survey results show that many teachers have poor mental health, and that some have mild–moderate or even severe mental problems [1,2,3,4]. Teachers are the key to education, and their well-being directly affects their work efficiency and directly or indirectly affects students’ learning and health [5,6]. Primary and secondary school students are in especially critical periods of personality-building, and their psychological status and personality development are easily affected [7]. Thus, the well-being of teachers warrants attention, not only to improve the quality of basic education but also to meet students’ need for a well-rounded education. The present research aims to shed light on the relationship between work stress and the well-being of primary and secondary school teachers, with the proposal of a new model including potential moderators (e.g., self-transcendent meaning of life), a mediator (work–family conflict), and the provision of evidence supporting the model.

### 1.1. Work Stress and the Well-Being of Primary and Secondary School Teachers

Primary and secondary school teachers face long working hours as well as high labor intensity and stress levels [8]. Operating under such conditions for a long time affects their health status [9,10]. According to the conservation of resources (CoR) theory, work stress consumes individual resources (e.g., time, energy, psychological resources), eventually having many adverse effects [11]. Among teachers, these effects include mental and physical fatigue, nervous tension, frustration, torment, and other unpleasant negative emotional experiences [12] generating a series of adverse psychological, physiological, and behavioral reactions. Thus, we propose the following hypothesis:

**Hypothesis** **1.** 
*Teachers’ work stress predicts poor well-being.*


Researchers often use the 90-item Symptom Checklist to investigate teachers’ well-being [13,14]. In positive psychology, well-being is conceived of as not only the absence of depression, but also the fulfillment of the need for happiness [15]. As teachers interact with students at work, which requires a large amount of psychological resources [6,16], we used the Chinese version of the Short Depressive–Happiness Scale, revised by Wang et al. based on positive psychology, to measure teachers’ psychological well-being [15].

### 1.2. Family–Work Conflict as a Mediator

Family–work conflict was defined as a kind of role conflict, which is the conflict generated when individuals’ roles change between family and work and Greenhaus et al. believe that the conflict between family and work is bidirectional [17]. Among them, work–family conflict is directed from work to family, while family–work conflict is directed from family to work, which is the impact of family demands on work [18]. Although most researchers agree on the bidirectional nature of conflict, more researchers study the work-to-family conflict, and some studies show that the work-to-family conflict is more prevalent than family-to-work conflict [19]. However, with the development of society, people are attaching more importance to family. In China, Gong Huoliang [20], Wu Mingxia, Zhang Dajun [21] and other researchers have studied family-to-work conflict with other variables, such as depression and occupational behavior. Therefore, the purpose of this study is to explore family-to-work conflict (FWC) in order to explore how to prevent conflict and intervene for teachers’ well-being from the perspective of family.

Work and family are two important parts of a person’s life [22]. Teachers exhibit low degrees of psychological disengagement [9,23], as they must often continue working after coming home, which occupies time and leads to the inability to meet the needs of their families. According to the CoR theory, the allocation of individual resources to family and work is dynamic, but needs to be relatively stable; when its stability is disrupted, conflicts occur [11,24]. The inability to meet work needs due to family needs is an example of family–work conflict, high levels of which have been observed among teachers [25]. Such conflict has negative impacts on well-being [26]. Thus, we propose the following hypothesis:

**Hypothesis** **2.** 
*Family–work conflict mediates the association between teachers’ work stress and well-being.*


### 1.3. Self-Transcendent Meaning of Life as a Moderator

“Meaning” is an important source of well-being. Individuals with low levels of well-being tend to experience “meaninglessness” and “valuelessness” [27,28,29]. Yalom [30] defined transcendent and secular (self-centered) types of meaning. Although some research has explored relationships between the meaning of life and well-being in China and the West, it is not clear whether the “sense of self-transcendence” or the “sense of self-centeredness” is at work [27].

Reed considered self-transcendence to be defined as the ability to transcend personal concerns and have a broader perspective and purpose in life [31,32]. The definition of a self-transcendent meaning of life in this study is based on Chinese researcher Li Hong’s definition: the cognition and belief of a higher state of existence beyond the self (getting rid of self-centeredness). This definition emphasizes identification, belief, and awareness beyond the realm of self-existence (from a psychological point of view, it is essentially a cognitive and belief system, embodied in people’s attitudes toward life) [27]. It is considered to be a healing resource that confers well-being due to a heightened awareness of wholeness and the integration of all aspects of one’s being. Individuals with a self-transcendent meaning of life have entered a realm beyond the self, as conceived of in Chinese philosophy and culture [27,33]. The exploration of this conceptualization is an important direction of Chinese localization research. Buddhist philosophy holds that people’s suffering stems from excessive persistence. Taoist philosophy holds that everything in the world exists dialectically, and thus, people should let nature take its course and view gains and losses and success and failure equivocally while maintaining a normal state of mind [15,34], moving from “pain” to “rebirth” [35].

A self-transcendent perspective on the meaning of life can help individuals regulate the negative emotions caused by stressful events [27,36,37,38,39,40] through a detached state of mind resulting from a non-egocentric understanding of their own value [41]. Psychological health problems are related not only to stressful events, but also individuals’ perspectives on their own value. Research has shown that individuals with high levels of self-transcendence have better health status due to their ability to resolve negative emotions [33,42,43,44]. Thus, we propose the following hypothesis:

**Hypothesis** **3.** 
*A self-transcendent meaning of life moderates the impact of teachers’ work stress on their well-being.*


### 1.4. The Present Study

As a collectivist country, family is an important source of individual happiness in China, so the family is also crucial to individual happiness [45]. The self-transcendent meaning of life in Eastern culture and Western culture may be different. This study is the first to explore the impact of a self-transcendent meaning of life based on Chinese traditional culture on teachers’ well-being. The results of this study suggest a new way for educators and related parties to intervene to improve the well-being of primary and secondary school teachers in a manner consistent with traditional Chinese culture. Based on our literature review, we hypothesized that this relationship would be mediated by work–family conflict and moderated by a self-transcendent meaning of life (Figure 1). We tested our hypotheses among primary and secondary school teachers in China.

## 2. Method

### 2.1. Participants

We recruited seven researchers from Jiangxi Normal University and trained them uniformly. A total of 600 questionnaires were distribute in several primary and secondary schools in Jiangxi and Yunnan provinces, China, in November 2018. The subjects in Jiangxi Province were primary and secondary school teachers participating in the national training program (a continuing education training required for primary and secondary school teachers in China). A total of 365 teachers from Jiangxi province were recruited and valid completed questionnaires were received from 353 of them, after exclusion of questionnaires that were unfifinished or that had garbled responses. Additionally, 235 teachers from Yunnan Province were recruited and valid data were obtained from 204 of them.Participants filled out the questionnaire anonymously and were assured that their responses would remain confidential. After the exclusion of questionnaires with overly regular answers and missing answers, the sample comprised 562 valid questionnaires, 455 from primary schools (grades 1–6) and 99 from secondary schools (grades 7–12); 8 people did not fill out this column. The sample comprised 279 (49.6%) men and 283 (50.4%) women with a mean age of 38.44 (min = 20, max = 60, *SD* = 10.51) years. The sex ratio was similar to previous surveys [46,47]. This study scheme was approved by the Institutional Review Committee of Jiangxi Normal University, Nanchang, China (IRB-JXNU-PSY-20180019). All methods were performed in accordance with relevant guidelines and regulations.

### 2.2. Measures

#### 2.2.1. Work Stress

Work stress was measured using items from the Work-Pressure Questionnaires of Teachers in Primary and Middle Schools [48]. The original scale consists of two parts examining stressors and stress responses, respectively. We selected 12 items related to student problems and job characteristics from the stressors subscale, as these aspects have been shown to be the main sources of pressure for teachers in China [49]. Responses are structured by a 5-point Likert scale (0 = no stress, 4 = a lot of stress). The Cronbach’s *α* value for these items was 0.93.

#### 2.2.2. Work–Family Conflict

We measured family–work conflict using the family-to-work subscale of the Work–Family Conflict Scale developed by Netemeyer et al. [50]. This subscale consists of 5 items that describe conflicts in which family responsibilities interfere with work responsibilities. Participants were asked to indicate on a 7-point Likert scale (1 = strongly disagree, 7 = strongly agree) whether the statements described their experiences. Higher scores reflect more serious conflict. The Cronbach’s *α* value for this subscale was 0.90.

#### 2.2.3. Well-Being

We used the Chinese version of the Short Depression–Happiness Scale [15] revised by Wang et al. to evaluate teachers’ well-being in the past week in terms of depression and happiness. The scale consists of 6 items with responses ranging from 1 (never) to 4 (often). After the reversal of the response scale for 3 items, higher total scores reflect better well-being. The Cronbach’s *α* value for this scale was 0.82.

#### 2.2.4. Self-Transcendent Meaning of Life

We applied the Self-Transcendence Meaning of Life Scale (SMLS) compiled by Li [36] and revised by Wang and Jing [28]. Wang and Jing explored two factors. One is “grasping the meaning of failure” (GMF), based on the Taoist concepts of gain and loss, and the other is “detachment from success or failure” (DSF), based on the Buddhist concept of “making the ego fade away.” We explored these distinct philosophical concepts separately in this study. This 8-item scale measures GMF and DSF with items such as “success and failure have positive meanings for people.” Responses range from 1 (strongly disagree) to 4 (strongly agree). Higher scores reflect greater self-transcendence in understanding the meaning of life. The Cronbach’s *α* value for this scale was 0.90.

### 2.3. Statistical Analysis

The data were analyzed using SPSS 23.0 and the PROCESS macro package. First, we performed common variance analysis to detect common method biases. Second, we calculated descriptive statistics, created a correlation matrix, and did a difference analysis. Third, we used the PROCESS macro (model 4) developed by Hayes [51] to test for the mediating effect of family–work conflict, controlled for respondent sex, age, and type of school. Finally, we ran model 5 with bootstrapping to examine whether the self-transcendent meaning of life had a moderating effect [51]. Percentile bootstrap estimation with 5000 resamples was used to generate 95% confidence intervals (CIs). CIs not including 0 were considered to reflect significant effects.

## 3. Results

### 3.1. Common Method Bias

As all study data were self-reported, they may be affected by common method deviation. Corresponding control measures (anonymization and reverse scoring of some questions) were taken. However, to ensure the rigor of the results, we used Harman’s single-factor test to identify common method deviation. The first factor explained 29.10% (less than the 40% standard) of the variance, indicating that no serious common method bias affected the study data.

### 3.2. Descriptive Statistics and Difference Analysis

Teachers’ work stress correlated positively with family–work conflict and negatively with well-being and SMLS total and factor scores (all *p* < 0.001; Table 1). Family–work conflict was correlated negatively with well-being and SMLS total and factor scores (all *p* < 0.001). Well-being correlated positively with SMLS total and factor scores, and total SMLS scores correlated positively with the two factor scores of this scale (all *p* < 0.001; Table 1).

There were significant gender differences in work stress, family–work conflict, SMLS, and factor-DSF. Women perceived a higher level of work stress and family–work conflict, but a lower level of self-transcendent meaning of life. There was no significant gender difference in well-being. There were significant differences in work stress, well-being, family–work conflict, and factor-GMF in primary school and secondary school. Teachers in secondary schools perceived higher level of work stress and family–work conflict, and lower level of well-being and GMF (Table 2).

### 3.3. Mediating Effect of Family–Work Conflict

Teachers’ work stress significantly predicted family–work conflict [*β* = 0.21, standard error (SE) = 0.03, *p* < 0.001]. Well-being was significantly predicted by teachers’ work stress (*β* = −0.16, SE = 0.01, *p* < 0.001) and family-to-work conflict (*β* = −0.06, SE = 0.02, *p* < 0.01). Bootstrap testing showed that the mediating effect of work–family conflict was significant (*β* = −0.01, bootstrap SE = 0.005; 95% CI, −0.022 to −0.003) and accounted for 6.83% of the total effect.

### 3.4. Moderating Effects of the Self-Transcendent Meaning of Life, GMF, and DSF

SMLS scores positively predicted well-being (*β* = 0.12, *p* < 0.001; bootstrap interval, 0.057–0.188), and the interaction between work stress and SMLS scores negatively predicted well-being (*β* = −0.01, *p* < 0.05). A simple slope test showed that work stress significantly affected the well-being of teachers with low SMLS scores (*Z* = −1; simple slope = −0.12, *t* = −6.38, *p* < 0.001), but had a more significant effect for teachers with high SMLS scores (*Z* = 1; simple slope = −0.17, *t* = −11.60, *p* < 0.001; Figure 2, Table 3 and Table 4). However, the plot shows that the well-being of teachers with high SMLS scores was always greater than that of teachers with low SMLS scores under different levels of work stress. These findings reflect the protective effect of a self-transcendent meaning of life and its weakening with increasing work stress, consistent with the “protection-response” regulation model [52].

GMF scores positively predicted well-being (*β* = 0.18, *p* < 0.01; bootstrap interval, 0.083–0.283). The interaction of work stress with GMF negatively predicted well-being (*β* = –0.01, *p* < 0.05; Table 2). A simple slope test showed that work stress significantly affected the well-being of teachers with low GMF scores, but that the effect was more significant for teachers with high GMF scores (Figure 3, Table 2 and Table 3). The plot shows that the well-being of teachers with high GMF scores was greater than that of teachers with low GMF scores under different levels of work stress, but the protective effect of GMF weakened with increasing work stress.

DSF scores positively predicted well-being (*β* = 0.24, *p* < 0.01; bootstrap interval, 0.080–0.406). The interaction of work stress with DSF negatively predicted well-being (*β* = −0.01, *p* < 0.05; Table 2). A simple slope test showed that work stress significantly affected the well-being of teachers with low DSF scores, but that the effect was more significant for teachers with high DSF scores (Figure 4, Table 2 and Table 3). The plot shows that the well-being of teachers with high DSF scores was greater than that of teachers with low DSF scores under different levels of work stress, but the protective effect also weakened with increasing work stress.

## 4. Discussion

In the difference analysis, we found that males’ SMLS and factor-GMF scores were significantly higher than that of females. It may be due to the fact that school teaching is a female-dominated profession [53]. Sociocultural biases and low salaries lead to male teachers in primary and secondary schools having lower professional identity than females [54], and male teachers find it more difficult to obtain social support [55]. In the face of this situation, male teachers are more likely to relieve pressure through self-guidance and self-transcendence, taking success and failure in stride. As was said, self-transcendence could be a vital resource for the well-being of vulnerable populations [56]. In addition, most of the archetypes and related stories of self-transcendence that are praised and embodied in Chinese traditional culture are males (such as Liang Qichao, Su Dongpo, etc.) [57,58], while female archetypes and stories are extremely rare. In this cultural and historical background, men may have more recognition and internalization of the self-transcendent meaning of life than women. The scores of primary school teachers in factor-GMF were significantly higher than those of secondary school teachers. It may be due to the fact that primary school teachers have less teaching tasks and lower social expectations. They are worse than middle school teachers in terms of salary and social status. However, primary school teachers are staff of public institutions, as are middle school teachers, which means their jobs are stable. In addition, according to the statistics of the Bureau of Education (http://www.moe.gov.cn/), the educational level of teachers who work in primary schools is lower than that of teachers working in secondary schools. Teachers in secondary schools may have higher requirements and expectations for themselves, so they have lower tolerance for failure.

Work stress negatively predicted the well-being of primary and secondary school teachers in this study, supporting Hypothesis 1, which is consistent with previous findings [36,59,60]. Teachers play a very important role associated with high levels of work stress. In China, teachers are no longer the authority figures that they once were; they need to be responsible for their every word and behavior, and are subject to social supervision [61]. Teachers must complete numerous tasks, which requires substantial amounts of psychological resources, time, and energy. Well-being is affected by many individual and environmental factors, and is compromised by excessive work stress with which individuals cannot effectively cope [62].

We found that family–work conflict partially mediated the impact of work stress on the well-being of primary and secondary school teachers; that is, work stress leads to family–work conflict, which reduces well-being. Thus, our data support Hypothesis 2. As a collectivist country, family is one of the important sources of individual happiness in China; people are increasingly aware of the importance of family roles, and family needs are increasing [45]. According to the CoR theory, the allocation of limited individual resources to family and work entails checks and balances [11]. Excessive work stress requires more resources to be called, and family-to-work conflict occurs when individuals are unable to call resources from their families. According to boundary theory, there should be not only a boundary, but also dynamic equilibrium between family and work [20]. When work stress is excessive and individuals must delay work because of family needs, this conflict may be internalized as guilt and/or remorse, likely leading to emotional exhaustion [63] and reduced well-being. In addition, the role theory holds that the resolution of family–work conflict requires the changing of roles between the two arenas in a recovery effort that consumes a lot of resources.

A self-transcendent meaning of life moderated the effect of work stress on the well-being of primary and secondary school teachers in this study. This perspective protected the teachers’ well-being; the greater well-being of teachers with high SMLS scores than that of those with low SMLS scores is consistent with previous findings [33,42,43,44]. However, this protective effect weakened with increasing work stress. A self-transcendent meaning of life emphasizes letting go of obsession and treating success and failure dialectically [26]. Individuals with such concepts have different cognitions and beliefs regarding stressful events [27]. This perspective can relieve pressure [37,38] and enable the effective regulation of emotions [64].

It is difficult for teachers to be all things to all people at once. In this case, one must “let nature take its course” and “do one’s best.” Teachers should face failure with a positive attitude and gain experience from it. According to self-worth theory, individuals with high drive and avoidance levels center their self-worth and conceptualization of the meaning of life on the achievement of success and avoidance of failure; the excessive desire for success and fear of failure aggravate their perception of pressure [65]. The weakening of the protective effect from the self-transcendent meaning of life with increasing work stress may be due to the conflict between these individuals’ values (i.e., “make the ego fade away” and “let nature take its course”) and the meaning that they pursue, which produces unpleasant experiences. Therefore, we can start with Chinese traditional culture to intervene in teachers’ mental health problems, such as the Chinese Taoist cognitive therapy developed by Yang Desen et al. [66], which was shown to be effective in a study of Chinese American immigrants [34].

GMF and DSF both moderated the relationship between teachers’ work stress and well-being in this study, but this effect weakened with increasing work stress. There are different subcultures in traditional Chinese philosophy, some of which advocate letting nature take its course and maintaining a peaceful state of mind—as advocated by Taoist cognitive therapy—which has a positive effect on depression, anxiety, and other problems [34,67]. There are also views that excessive obsession is the root cause of psychological problems, and the solution is to detach from the obsession with success or failure [68]. Although these cultures are different, they all help individuals achieve self-transcendence.

### 4.1. Implications

The results of this study suggest a new way for educators and related parties to intervene in improving the well-being of primary and secondary school teachers in a manner consistent with traditional Chinese culture. First, as work stress directly affects teachers’ well-being with adverse psychological effects, the administrators of departments and schools should pay attention to teachers’ work burdens and take action to reduce them when appropriate. School administrators should create good working environments that reduce teachers’ stress. Second, teachers should pay attention to family–work balance and allocate time reasonably to reduce the occurrence of conflicts. When teachers have negative emotions, family members should come to their aid in a timely manner, creating a harmonious family atmosphere and giving teachers support to increase their psychological resources. Finally, to prevent well-being problems among teachers, school administrators should offer cultural activities and/or courses based on Chinese cultural and philosophical perspectives, training teachers to “let nature take its course” and “make the ego fade away,” have a dialectical view of gains and losses, and find meaning in failure. However, given the weakening effects of such perspectives with increasing work stress, the control of primary and secondary school teachers’ work stress is the root of solving the problem.

### 4.2. Limitations and Prospects

Several limitations of this study should be recognized. First, because of its cross-sectional rather than longitudinal design, this study could not provide evidence for the long-term efficacy of the moderating effect of a self-transcendent meaning of life. Second, this study was a questionnaire-based survey; future research could involve interventions targeting participants’ level of self-transcendent meaning of life to better test this factor’s moderating effect. Third, we explored primary and secondary school teachers at the same time. Although the type of school is controlled in the model, subsequent researchers can study independently in primary or secondary schools in order to provide fitting and specific suggestions. Finally, this study was performed in the Chinese educational context; similar studies could be conducted in other countries.

## 5. Conclusions

The present study was conducted to analyze the influence of work stress on the well-being of teachers in China, and to explore the effects of family–work conflict and a self-transcendent meaning of life. The results show that work stress negatively predicts the psychological well-being of primary and secondary school teachers; family–work conflict played a mediating role in the impact of work stress on the well-being of primary and secondary school teachers, with a mediating effect of −0.01, accounting for 6.85% of the total effect; self-transcendent meaning of life plays a moderating role in the impact of work stress on the well-being of primary and secondary school teachers, and with an increase in work stress, the protective role of a self-transcendent of meaning of life is weakened; the moderating effect of factor-GMF and factor-DSF was significant.

## Figures and Tables

**Figure 1 ijerph-20-01154-f001:**
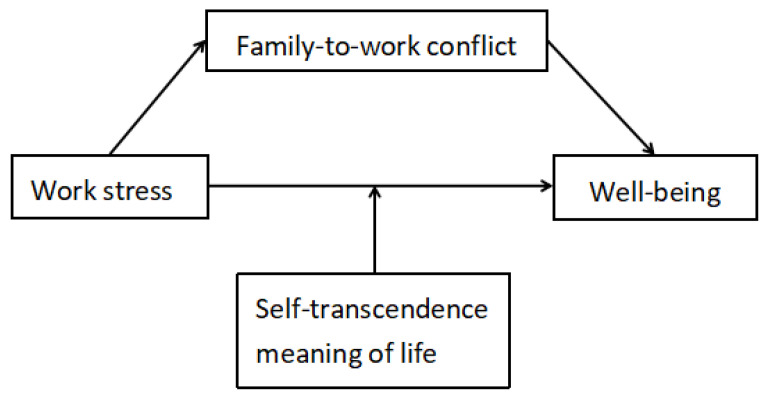
Hypothesis model.

**Figure 2 ijerph-20-01154-f002:**
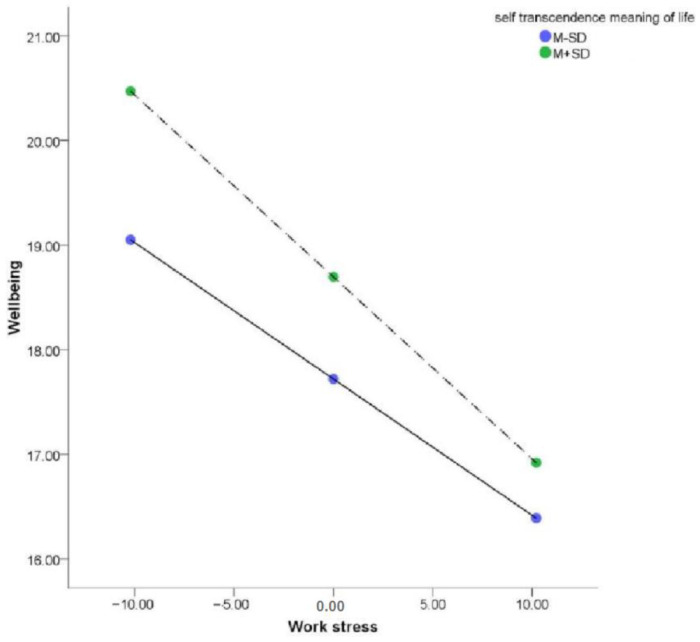
The moderating effect of a self-transcendent meaning of life.

**Figure 3 ijerph-20-01154-f003:**
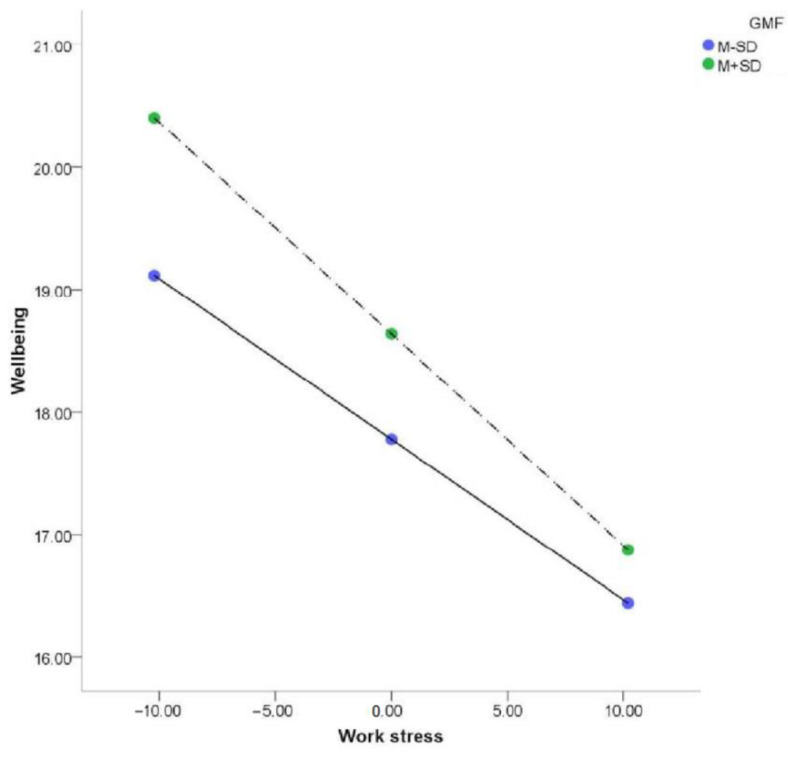
The moderating effect of factor-GMF (grasping the meaning of failure).

**Figure 4 ijerph-20-01154-f004:**
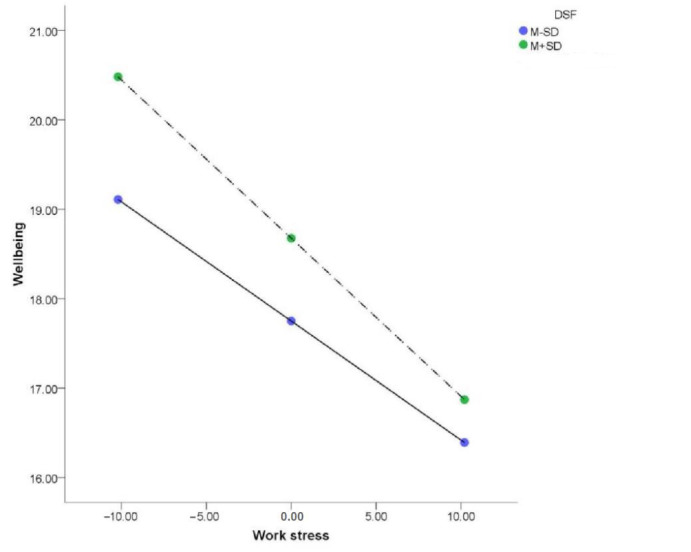
The moderating effect of factor-DSF (detachment from success or failure).

**Table 1 ijerph-20-01154-t001:** Descriptive statistics and correlation of variables (*n* = 562), Pearson correlation coefficients of all variables.

	*M* ± *SD*	Age	1	2	3	4	5	6
1. Work stress	27.01 ± 10.21	0.152 ***	1					
2. Family-to-work conflict	12.14 ± 6.86	−0.082	0.306 ***	1				
3. well-being	18.25 ± 3.55	−0.017	−0.505 ***	−0.262 ***	1			
4. SMLS	24.74 ± 4.15	0.016	−0.179 ***	−0.203 ***	0.238 ***	1		
5. Factor-GMF	8.81 ± 1.79	0.000	−0.184 ***	−0.167 ***	0.226 ***	0.890 ***	1	
6. Factor-DSF	15.94 ± 2.68	0.025	−0.155 ***	−0.202 ***	0.218 ***	0.953 ***	0.710 ***	1

Note: *** *p* < 0.001.

**Table 2 ijerph-20-01154-t002:** Difference analysis of gender and type of school.

	Sex			Type of School (Primary/Secondary)		
	*M* (Men/Women)	*t*	*p*	*M* (Primary/Secondary)	*t*	*p*
1. Work stress	25.52/28.49	3.50 **	<0.01	26.60/29.34	−2.44 *	<0.05
2. Family-to-work conflict	11.42/12.85	2.49 *	<0.05	11.89/13/38	−1.97 *	<0.05
3. well-being	18.50/18.00	−1.67	>0.05	18.42/17.29	2.88 **	<0.01
4. SMLS	25.16/24.33	−2.39 *	<0.05	24.87/24.22	1.41	>0.05
5. Factor-GMF	8.95/8.67	−1.85	>0.05	8.90/8.41	2.47 *	<0.05
6. Factor-DSF	16.22/15.66	−2.46 *	<0.05	15.97/15.81	0.53	>0.05

Note: * *p* < 0.05, ** *p* < 0.01.

**Table 3 ijerph-20-01154-t003:** The moderating effect of self-transcendent meaning of life, factor-DSF, and factor-GMF.

Outcome Variable	Regulated Variable	Predictive Variable	*β*	BootLLCI	BootULCI	*t*	*R* ^2^	*F*
Well-being	self-transcendent meaning of life	Work stress	−0.15	−0.176	−0.118	−10.52 ***	0.29	32.44 ***
		Family-to-work conflict	−0.04	−0.087	−0.002	−2.21 *		
		Work stress * SMLS	−0.01	−0.011	−0.001	−2.24 *		
	Factor-DSF	Work stress	−0.15	−0.182	−0.127	−10.42 ***	0.29	31.73 ***
		Family-to-work conflict	−0.05	−0.089	−0.004	−2.40 *		
		Work stress * Factor-DSF	−0.01	−0.015	>−0.001	−2.07 *		
		Work stress	−0.15	−0.178	−0.121	−10.81 ***	0.29	32.11 ***
	Factor-GMF	Family-to-work conflict	−0.04	−0.087	−0.002	−2.43 *		
		Work stress * Factor-GMF	−0.01	−0.016	−0.001	−2.21 *		

Note: Covariates controlled for the effects of gender and age. * *p* < 0.05, *** *p* < 0.001.

**Table 4 ijerph-20-01154-t004:** The moderating effect of self-transcendent meaning of life, factor-DSF, and factor-GMF on path.

Moderating Variable	Level of Factor	Path	Effect Size	LLCI	ULCI
self-transcendence meaning of life	*M* − *SD*	X→Y	−0.12	−0.163	−0.086
	*M*	X→Y	−0.15	−0.175	−0.120
	*M + SD*	X→Y	−0.17	−0.201	−0.141
Factor-DSF	*M* − *SD*	X→Y	−0.13	−0.165	−0.086
	*M*	X→Y	−0.15	−0.176	−0.120
	*M + SD*	X→Y	−0.17	−0.200	−0.140
	*M* − *SD*	X→Y	−0.13	−0.165	−0.089
Factor-GMF	*M*	X→Y	−0.15	−0.178	−0.123
	*M + SD*	X→Y	−0.17	−0.205	−0.144

Note: X is “work stress”, Y is “well-being”.

## Data Availability

Some or all data, models, or code generated or used during the study are available from the corresponding author by request.

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
