# Peer review of "The Effect of Work Stress on the Well-Being of Primary and Secondary School Teachers in China"

_ijerph, 2023, doi:10.3390/ijerph20021154_

Round 1

Reviewer 1 Report

The title is promising, with meaningful insights for the targeted areas with a proper methodology.

Nevertheless, there are some basic problems. The introduction lists some sources proving the importance of the very basic concepts and elements of the present research, but actually, those are not highly prominent and proven pieces of literature, nor show the adequateness of the concepts (e.g., 6-8). In the case of 9, the German approach is proved, but not the Chinese situation, which therefore is not sufficiently justifying the basic statement of the overwhelming working hours of the teachers. Going further, it is evidenced that the introduction should play the role of the literature review, which is not fulfilling its purpose. The hypotheses are not underlined and proved properly. Similar issues are to be found for the other cases, too, with an improper and inadequate presentation of the basic concepts.

The sampling methodology is not detailed enough, there is no information about the basic population, and the validity of the sample is neither ensured. Regarding the questionnaire, only the work-family conflict-related elements can be accepted without further concerns about their validity; the other elements are not well-elaborated and widely controlled.

The statistical method could be interesting, but there is no justification why for using the model 5, why actually the identified elements are mediating or moderating. Due to these uncertainties, the professional content of the article cannot be evaluated and considered acceptable for the purpose.

The implications are rather unuseful for the targeted stakeholders without a clear recommendation of how they exactly could be used as reference points or guidelines.

The final conclusions are rather confusing because of their non-embeddedness in the normal structure. 

Author Response

Thank you for your comments and suggestions! Please see the attachment.

Reviewer 2 Report

I don't think the authors used the template jurnal in the last version - there are no line numbers.

Title:

I am not convinced that the title must indicate the statistical methods used.

maybe it is just enough: The effect of work stress on the well-being of primary- and sec-ondary-school teachers in China.

Abstract:

Also in the abstract, I would avoid indicating direct statistical results e.g. [β = –0.17, standard error (SE) = 0.01, p <0.001). Such presentation of data is very rarely practiced. The mere presentation of the results is sufficient without the need to present the obtained statistical results

 Keywords:

One of the tasks of keywords is to enable other researchers to find research articles of interest to tchem. I highly doubt that such keywords would be used by any of the researchers: primary- and secondary-school teachers,   self-transcendent conceptualization of the meaning of life. Please choose keywords so that other authors can easily find this article.

 Introduction

the footnotes in this Journal are used in square brackets but are not written superscript at all.

The introduction contains the most important information but appears to be disordered. I suggest the authors read this text at a distance once again and make a consistent presentation of the information.

Participants

Do the gender ratios in the study match the proportion of the general teacher population in China?

Please indicate the date on which the survey was conducted.

Measures

The research tools presented have been correctly described and there is no need to introduce changes here.

Results

What kind of correlation was used to present the data in table one? Pearson correlation?

Please remember that the correlation with the use of dichotomous variables such as gender should be used Point-two-series correlation coefficient

In articles of this kind, we do not repeat statistical data in the text if they have already appeared in the table

Discussion

Both the discussion and the conclusion present the most important information. The reader is well informed about the results obtained in the context of other studies.

the bibliography is not prepared according to journal standards

Author Response

(The authors gave the same response as above.)

Reviewer 3 Report

1. Please correct the typos in the abstract "cluster sampling method,,"

2. In the abstract you can avoid mentioning the dimensions again, you have already explained that in the previous line. It is not usual to list the results with betas and p-values in the abstract. It would be beneficial if you could instead describe the results and present the main conclusions from your study.

3. In the introduction, it says that teachers' mental health suffers in different regions, but you only mentioned two studies, one from a Polish journal and one from China. Would it be helpful for the study to include more sound and solid research on this topic?

4. It is not entirely clear why you included the concept of self-transcendence of life in your study 

5. "Although some research has examined the relationships between the meaning of life and well-being in China and the West, the concept of the meaning of life is relatively vague, and the influence of a self-transcendent conceptualization of this meaning is rarely explored independently [6]." Could you please elaborate on this point?

6. You said that researchers often use the 90-item Symposium Checklist to study teachers' well-being, but you mentioned only one study.

7. You should explain in a little more detail what family-work conflict is and why it is different from work-family conflict. It is also not clear how you justify your hypothesis. If teachers are burdened with their workload and often have to continue working after they get home (is there any evidence to support this statement?), does the inability to meet the needs of the family interfere with the work-family relationship or vice versa? In other words, are you sure you are considering the family-work conflict or the opposite? The mediation hypothesis is very concise and hardly supported by previous research studies. Please elaborate on this point.

8The moderation hypothesis could also be further supported by a detailed explanation of the construct. It should be clear whether it is a philosophical construct, a coping strategy from positive psychology or something else. Please explain this dimension and the two sub-concepts that are only very briefly described in the introduction 

9. Does West culture" refer to occidental culture? Should it be Western culture and not "the West"? This could be misleading if not clearly defined.

10. Did you ask the participants to indicate if they have had any experience with the philosophical construct/ or the coping strategy? 

11. If it is a way of seeing and making sense of our existence, why would the concept only moderate the relationship between work and well-being and not the other two paths? Please elaborate on this point.

12. In the Results section, the authors titled one of the paragraphs as "moderating effect of the self-transcendent conceptualization of the meaning of life, GMF and DSF", but in Tables 2 and 3, GMF is not included 

13. An index of the moderating mediation model should be estimated and reported in the paper to check whether the overall model is significant.

14. The statement that school administrators should create a good working environment that reduces teacher stress is generally quite obvious. The authors should demonstrate their skills in interpreting their findings and describe a more practical and effective way to support school administrators. Perhaps the work stress scale item could help with this... just a suggestion.

15. It might be helpful to discuss practical ways of bringing training activities to teachers based on the self-transcending conceptualisation of the meaning of life. The cultural openness of the organisation, in this sense, should also be considered.

Author Response

(The authors gave the same response as above.)

Reviewer 4 Report

This manuscript is interesting and innovative. It may also have practical significance. However, it needs some clarifications and additions.

Methods section doesn't contain the study ethics. It must be properly addressed and described in the manuscript (e.g., ethical committee approval, participants consents, design).

There is also a lack of more detailed data on the study group (for total group and divided by gender), work characteristics, number of hours spent in school and on work outside of school. Unclear to me is the statement: „Using a cluster sampling method, we recruited seven researchers who were trained uniformly to distribute 600 questionnaires in several primary and secondary schools in Jiangxi and Yunnan provinces, China”. Were the 7 researchers,  selected using this metod, or the schools in which the surveys were conducted? Why the Authors decided to perform survey in 600 teachers? Is it such  group represenatative for teachers in these two Provinces?

The description of the methods lacks information on what the range of results obtained might be (min-max.). Consequently, the M±SD data in Table 1 tell us nothing. Are these results high, medium, or low?  Because of the significant difference regarding gender on "Work stress" and "Family-to-work conflict," it would be worthwhile to conduct analyses separately for both genders.  The mere inclusion of gender as a covariate in other analyses is insufficient in my opinion. The influence of school type (primary/secondary) was also not included in the analyses. Based on my own research, I can conclude that it is significant.

In the Introduction and Disscussion, the Authors refer primarily to data from the Asian literature; there are too few literature references to studies from Europe and America.

Author Response

(The authors gave the same response as above.)

Round 2

Reviewer 1 Report

The applied changes are putting in focus the suggested corrections and now it is with some minor flaws, acceptable. 

Author Response

Thank you for your affirmation. Your suggestions are very useful to the article.

Reviewer 2 Report

Thank you to the authors for submitting a revised version of the article. The corrections made have significantly increased the quality of this article.

Author Response

(The authors gave the same response as above.)

Reviewer 3 Report

I thank the authors for responding to many of the comments on their study. The paper is improved and can be published.

I ask the authors to take care of a few last open points.

1. The authors should clarify what GMF and DSF mean. The first time these words appear in the text, no translation of the acronym is given. Please correct this.

2. The PROCESS package also provides the moderate mediation index in the output dialogue box. Please specify it in the Results section and comment accordingly 

3. I still have doubts about your concept of family-work conflict when you state the following: "Excessive work pressure occupies resources and makes the achievement of balance, leading to family-work conflict". In this sense, it seems to be more of a work-family conflict than the other way around 

4. It would be appropriate to comment on two types of findings in the discussion: significant differences in terms of gender and comment on the differences between GMF and DSF between primary and secondary school. Why, according to the authors, do these differences stand out? Please discuss.

Author Response

Thank you very much. Your suggestions are very useful to the article. I have revised the article. Please see the attachment.

Reviewer 4 Report

I thank the Authors for their reply and corrections, but I still have reservations about the description of the method. What were the "cluster sampling methods" (one stage, two stage, multiple stage?). How many schools were part of the clusters. It should be discussed in details. How it was evaluated the number of questionnaires, why did you decide to distribute 600 questionnaires?

Regarding the cited literature, three recently added items that were supposed to concern teachers from other regions of the world than Asia are unavailable in PubMed, as is the doi for [1] Deng T., Shao Y. D., Ye M. S.&Li Y. (2022).

Therefore, I cannot check whether these items concern other than Chinese regions of the world, but the names of the authors show that rather not. Thus, the Authors of this manuscript have not made efforts to supplement the literature with research results from other regions of the world (Europe, America), and such items are available, e.g. Milić S, Marić N. Concerns and mental health of teachers from digitally underdeveloped countries regarding the reopening of schools after the first wave of the COVID-19 pandemic. Work. 2022;71(1):53-64. doi: 10.3233/WOR-210885. PMID: 34924427

Author Response

Thank you very much. Your opinions are very useful to us. I have revised the article according to your suggestions. Please see the attachment.
